# The Relationship between Fuzzy Reasoning Methods Based on Intuitionistic Fuzzy Sets and Interval-Valued Fuzzy Sets

**Minxia Luo \***  , **Wenling Li and Hongyan Shi**

Department of Date Science, China Jiliang University, Hangzhou 310018, China
\* Correspondence: mxluo@cjlu.edu.cn; Tel.: +86-571-8691-4480

**Abstract:** Two important basic inference models of fuzzy reasoning are Fuzzy Modus Ponens (FMP) and Fuzzy Modus Tollens (FMT). In order to solve FMP and FMT problems, the full implication triple I algorithm, the reverse triple I algorithm and the Subsethood Inference Subsethood (SIS for short) algorithm are proposed, respectively. Furthermore, the existing reasoning algorithms are extended to intuitionistic fuzzy sets and interval-valued fuzzy sets according to different needs. The purpose of this paper is to study the relationship between intuitionistic fuzzy reasoning algorithms and interval-valued fuzzy reasoning algorithms. It is proven that there is a bijection between the solutions of intuitionistic fuzzy triple I algorithm and the interval-valued fuzzy triple I algorithm. Then, there is a bijection between the solutions of intuitionistic fuzzy reverse triple I algorithm and the interval-valued fuzzy reverse triple I algorithm. At the same time, it is shown that there is also a bijection between the solutions of intuitionistic fuzzy SIS algorithm and interval-valued fuzzy SIS algorithm.

**Keywords:** Fuzzy Modus Ponens; Fuzzy Modus Tollens; reasoning algorithm; intuitionistic fuzzy sets; interval-valued fuzzy sets

**MSC:** 110.84

## 1. Introduction

In recent years, fuzzy control achieved great success in many aspects. Fuzzy reasoning is the core content of fuzzy control. As an important branch of approximate reasoning, fuzzy reasoning is close to human thinking mode. It has become the theoretical basis for fuzzy expert systems, fuzzy control systems and fuzzy intelligent decision systems, etc. In fuzzy reasoning, the most basic forms of fuzzy reasoning are Fuzzy Modus Ponens (FMP) and Fuzzy Modus Tollens (FMT) [1] as follows:

FMP: Given the input "$x$ is $A^*$", and fuzzy rule "if $x$ is $A$ then $y$ is $B$", try to infer a reasonable output "$y$ is $B^*$";

FMT: Given the input "$y$ is $B^*$", and fuzzy rule "if $x$ is $A$ then $y$ is $B$", try to infer a reasonable output "$x$ is $A^*$".

Zadeh [2] proposed the compositional rules of inference (CRI method for short) to deal with the above problem. Nevertheless, Wang [3] pointed out that the CRI method lacks strict logical basis and has no reducibility. Moreover, Wang [3] proposed full implication triple I method (triple I method for short), which improves the traditional CRI algorithm and brings fuzzy reasoning within the framework of logical semantic implication. Many researchers have done a lot of research on the triple I method and achieved a series of results. Wang and Fu [4] provided the unified forms of triple I method for FMP and FMT. Pei [5] comprehensively discussed the method based on residual fuzzy implication induced by left-continuous t-norms. Song and Wu [6] proposed a reverse triple I algorithm from the perspective of how to design a fuzzy system to minimize the number of elements in the fuzzy rule base under a given precision. Liu and Wang [7] proposed triple I method based on pointwise sustaining degrees. Luo and Yao [8] studied triple I algorithms based on

Schweizer–Sklar operators in fuzzy reasoning. In addition, the reducibility of the algorithm is one of the important criteria to evaluate the quality of fuzzy reasoning. Although the triple I algorithm and the reverse triple I algorithm have better properties in reducibility than CRI algorithm, their reducibility is not unconditional. Therefore, Zou and Pei [9] gave an SIS algorithm with the advantage of unconditional reducibility.

Although fuzzy sets have been successfully used in many fields, there are still some defects in describing the fuzziness and uncertainty of information. An interval-valued fuzzy set was introduced by Zadeh [10]. Many researchers extended approximate inference to the interval-valued fuzzy sets. An approximate reasoning method based on the interval-valued fuzzy sets was proposed [11]. Li et al. [12] discussed the robustness of interval-valued CRI method. Liu and Li [13] studied the interval-valued fuzzy reasoning with multi-antecedent rules. Luo and Zhang [14] extended the fuzzy inference triple I principle on interval-valued fuzzy sets, and gave the interval-valued fuzzy inference full-implication method based on the associated t-norms. Luo and Wang [15] further studied interval-valued fuzzy reasoning full implication algorithms based on the t-representable t-norm. Li and Xie [16] investigated universal interval-valued fuzzy inference systems based on interval-valued implications. Luo et al. [17] discussed the robustness of reverse triple I algorithms based on interval-valued fuzzy sets. Wang et al. [18] combined the SIS algorithm with interval-valued fuzzy sets to give a generalized SIS algorithm based on interval fuzzy reasoning and study its robustness.

Another extension of fuzzy sets, intuitionistic fuzzy sets, were proposed by Atanassov [19]. Many research results based on intuitionistic fuzzy sets have been obtained. Deschrijver et al. [20] proposed the intuitionistic fuzzy t-norm and t-conorm. Cornelis et al. [21] studied the intuitionistic fuzzy reasoning CRI method. Zheng et al. [22] studied the intuitionistic fuzzy reasoning triple I method and $\alpha$-triple I method. Liu and Zheng [23] proposed the dual triple I method and the decomposition method for intuitionistic Fuzzy Modus Tollens, which improved the reductivity of triple I method for intuitionistic Fuzzy Modus Tollens. Peng [24] discussed the intuitionistic fuzzy reasoning reverse triple I algorithm and the reverse $\alpha$-triple I algorithm. The literature [25] extended the SIS algorithm to intuitionistic fuzzy sets and then gave an SIS algorithm based on intuitionistic fuzzy reasoning and discussed its continuity.

Although scholars have made some research results based on intuitionistic fuzzy sets and interval-valued fuzzy sets, the relationship between the results has not been studied. This is the research goal of this paper. The structure of this paper is as follows: some concepts for intuitionistic fuzzy sets and interval-valued fuzzy sets are reviewed in Section 2. In Section 3, we study the relationship between the intuitionistic fuzzy reasoning triple I algorithm and interval-valued fuzzy reasoning triple I algorithm, the relationship between the intuitionistic fuzzy reasoning reverse triple I algorithm and interval-valued fuzzy reasoning reverse triple I algorithm, the relationship between the intuitionistic fuzzy reasoning SIS algorithm and interval-valued fuzzy reasoning SIS algorithm. The conclusions are given in Section 4.

## 2. Preliminary

In this section, we review some concepts for intuitionistic fuzzy sets and interval-valued fuzzy sets, which will be used in the paper.

**Definition 1** ([26]). *An increasing, commutative, associative mapping $T : [0,1] \times [0,1] \to [0,1]$ is called a triangular norm (t-norm for short) if it satisfies $T(x,1) = x$ for any $x \in [0,1]$. An increasing, commutative, associative mapping $S : [0,1] \times [0,1] \to [0,1]$ is called a triangular conorm (t-conorm for short) if it satisfies $S(0,x) = x$ for any $x \in [0,1]$.*

**Definition 2** ([27]). *The residuated implication $R$ induced by left-continuous t-norm $T$ is defined by $R(a,b) = \sup \{x \in [0,1] \mid T(a,x) \leq b\}, \forall a,b \in [0,1]$.*

**Example 1** ([28]). *(1) The Godel implication ($R_G$ for short) and the corresponding t-norm ($T_G$ for short) have the following expression*

$$R_G(a,b) = \begin{cases} 1, & if \quad a \leq b, \\ b, & if \quad a > b. \end{cases}$$

$$T_G(a,b) = a \wedge b.$$

(2) *The Lukasiewicz implication ($R_{Lu}$ for short) and the corresponding t-norm ($T_{Lu}$ for short) have the following expression*

$$R_{Lu}(a,b) = (1 - a + b) \wedge 1.$$

$$T_{Lu}(a,b) = (a + b - 1) \vee 0.$$

(3) *The Gougen implication ($R_{Go}$ for short) and the corresponding Product t-norm ($T_{Go}$ for short) have the following expression*

$$R_{Go}(a,b) = \begin{cases} 1, & if \quad a \leq b, \\ \frac{b}{a}, & if \quad a > b. \end{cases}$$

$$T_{Go}(a,b) = ab.$$

**Definition 3** ([19]). *An intuitionistic fuzzy set (IFS for Short) on nonempty universe X is given by*

$$A = \{(x, \mu_A(x), \vartheta_A(x)) \mid x \in X\}$$

*where $\mu_A(x) \in [0,1]$ and $\vartheta_A(x) \in [0,1]$ with the condition $0 \leq \mu_A(x) + \vartheta_A(x) \leq 1 (\forall x \in X)$. $\mu_A(x)$ and $\vartheta_A(x)$ are called a membership function and a non-membership function, respectively.*

The class of all intuitionistic fuzzy sets on nonempty universe $X$ is denoted $IFS(X)$.
For every $A, B \in IFS(X)$, some operations are defined as follows [29]:

(1)    $A \subseteq_{L^*} B$ iff $\mu_A(x) \leq \mu_B(x)$ and $\vartheta_A(x) \geq \vartheta_B(x), \forall x \in X$;
(2)    $A \cup_{L^*} B = \{(x, \sup(\mu_A(x), \mu_B(x)), \inf(\vartheta_A(x), \vartheta_B(x)) \mid x \in X\}$;
(3)    $A \cap_{L^*} B = \{(x, \inf(\mu_A(x), \mu_B(x)), \sup(\vartheta_A(x), \vartheta_B(x)) \mid x \in X\}$.

Let $L^* = \{(x_1, y_1) \mid (x_1, y_1) \subseteq [0,1]^2, x_1 + y_1 \leq 1\}$. The order defined on $L^*$ as $(x_1, y_1) \leq_{L^*} (x_2, y_2)$ if $x_1 \leq x_2$ and $y_1 \geq y_2$. $(x_1, y_1) \wedge_{L^*} (x_2, y_2) = (x_1 \wedge x_2, y_1 \vee y_2), (x_1, y_1) \vee_{L^*} (x_2, y_2) = (x_1 \vee x_2, y_1 \wedge y_2). \sup(x_i, y_i) = (\sup x_i, \inf y_i), \inf(x_i, y_i) = (\inf x_i, \sup y_i)$ for all $(x_i, y_i) \in L^*$. $0_* = (0,1)$ and $1_* = (1,0)$ are the smallest element and the greatest element in $L^*$, respectively. It is easy to verify that $(L^*, \wedge, \vee, 0_*, 1_*)$ is a complete lattice [30].

**Definition 4** ([20]). *An increasing, commutative, associative mapping $\mathcal{T}_{L^*}: L^* \times L^* \to L^*$ is called an intuitionistic fuzzy t-norm if it satisfies $\mathcal{T}_{L^*}(x, 1_*) = x$ for any $x \in L^*$.*

**Example 2** ([20]). *A binary mapping $\mathcal{T}_{L^*}: L^* \times L^* \to L^*$ is defined by $\mathcal{T}_{L^*}(\alpha, \beta) = (T(a_1, b_1), S(a_2, b_2))$, where $\alpha = (a_1, a_2), \beta = (b_1, b_2)$, S is the dual t-conorm of the t-norm T. Then, $\mathcal{T}_{L^*}$ is an intuitionistic t-norm, which is called the associated intuitionistic t-norm on $L^*$.*

*The associated intuitionistic t-norm $\mathcal{T}_{L^*}$ is called left-continuous if T is a left-continuous t-norm and S is a right-continuous t-conorm.*

**Definition 5** ([20]). *The intuitionistic residuated implication $\mathcal{R}_{L^*}$ induced by left-continuous intuitionistic t-norm $\mathcal{T}_{L^*}$ is defined by $\mathcal{R}_{L^*}(\alpha, \beta) = \sup\{\eta \mid \mathcal{T}_{L^*}(\eta, \alpha) \leq \beta\}$, where $\alpha, \beta, \eta \in L^*$, and $\mathcal{T}_{L^*}$ is a t-norm on $L^*$.*

**Lemma 1** ([22]). *The intuitionistic residuated implication induced by left-continuous associated intuitionistic t-norm $\mathcal{T}_{L^*}$ is $\mathcal{R}_{L^*}(\alpha, \beta) = (R(a_1, b_1) \wedge R(1 - a_2, 1 - b_2), 1 - R(1 - a_2, 1 - b_2))$, where $\alpha = (a_1, a_2), \beta = (b_1, b_2) \in L^*$, and R is the residuated implication induced by the t-norm T.*

**Definition 6** ([10]). *An interval-valued fuzzy set (IVFS for short) on nonempty universe X is given by*

$$B = \{(x, [B_l(x), B_r(x)]) \mid [B_l(x), B_r(x)] \subseteq [0,1], x \in X\}$$

The class of all interval-valued fuzzy sets on the nonempty universe $X$ is denoted $IVFS(X)$.

For every $A, B \in IVFS(X)$, some operations are defined as follows [10]:

(1)  $A \subseteq_{L^I} B$ iff $A_l(x) \leq B_l(x)$ and $A_r(x) \leq B_r(x), \forall x \in X$;

(2)  $A \cup_{L^I} B = \{(x, [\sup(A_l(x), B_l(x)), \sup(A_r(x), B_r(x))]) \mid x \in X\}$;

(3)  $A \cap_{L^I} B = \{(x, [\inf(A_l(x), B_l(x)), \inf(A_r(x), B_r(x))]) \mid x \in X\}$.

Let $L^I = \{[x_1, y_1] \mid [x_1, y_1] \subseteq [0, 1], x_1 \leq y_1\}$. The order defined on $L^I$ as $[x_1, y_1] \leq_{L^I} [x_2, y_2]$ if $x_1 \leq x_2$ and $y_1 \leq y_2$ is called component-wise order or Kulisch–Miranker order [31]. $[x_1, y_1] \wedge_{L^I} [x_2, y_2] = [x_1 \wedge x_2, y_1 \wedge y_2], [x_1, y_1] \vee_{L^I} [x_2, y_2] = [x_1 \vee x_2, y_1 \vee y_2]$. $\sup[x_i, y_i] = [\sup x_i, \sup y_i], \inf[x_i, y_i] = [\inf x_i, \inf y_i]$ for all $[x_i, y_i] \in L^I$. $0_I = [0, 0]$ and $1_I = [1, 1]$ are the smallest element and the greatest element in $L^I$, respectively. It is easy to verify that $(L^I, \wedge, \vee, 0_I, 1_I)$ is a complete lattice [31].

**Definition 7** ([32]). *An increasing, commutative, associative mapping $\mathcal{T}_{L^I} : L^I \times L^I \to L^I$ is called an interval-valued t-norm if it satisfies $\mathcal{T}_{L^I}(1_I, x) = x$ for any $x \in L^I$.*

**Example 3** ([33]). *A mapping $\mathcal{T}_{L^I} : L^I \times L^I \to L^I$ is defined by $\mathcal{T}_{L^I}(\alpha, \beta) = [T(a_1, b_1), T(a_2, b_2)]$, where $\alpha = [a_1, a_2], \beta = [b_1, b_2] \in L^I$, and T is a t-norm. Then, $\mathcal{T}_{L^I}$ is an interval-valued t-norm, which is called the associated interval-valued t-norm on $L^I$.*

*The associated t-norm $\mathcal{T}_{L^I}$ is called left-continuous if T is a left-continuous t-norm on the interval $[0, 1]$ [14].*

**Definition 8** ([20]). *The interval-valued residuated implication $\mathcal{R}_{L^I}$ induced by left-continuous interval-valued t-norm $\mathcal{T}_{L^I}$ is defined by $\mathcal{R}_{L^I}(\alpha, \beta) = \sup\{\mathrm{fl} \in L^I \mid \mathcal{T}_{L^I}(\mathrm{ff}, \mathrm{fl}) \leq \mathrm{fi}\}$.*

**Lemma 2** ([34]). *The interval-valued residuated implication induced by left-continuous associated t-norm $\mathcal{T}_{L^I}$ is $\mathcal{R}_{L^I}(\alpha, \beta) = [R(a_1, b_1) \wedge R(a_2, b_2), R(a_2, b_2)]$, where $\alpha = [a_1, a_2], \beta = [b_1, b_2] \in L^I$, and R is the residuated implication induced by the t-norm T.*

**Lemma 3** ([35]). *Mapping $\varphi : IFS(X) \to IVFS(X)$, $A \mapsto B$ is an isomorphism between the lattices $(IFS(X), \cup_{L^*}, \cap_{L^*})$ and $(IVFS(X), \cup_{L^I}, \cap_{L^I})$, where*

$$A = \{(x, \mu_A(x), \vartheta_A(x)) \mid x \in X\},$$

$$B = \{(x, [\mu_A(x), 1 - \vartheta_A(x)]) \mid x \in X\}.$$

## 3. The Relationship Based on Intuitionistic Fuzzy Sets and Interval-Valued Fuzzy Sets

*3.1. The Relationship between the Triple I Methods Based on Intuitionistic Fuzzy Sets and Interval-Valued Fuzzy Sets*

In this section, the relationship between the solutions of the triple I method based on the IFS and the IVFS will be studied.

**Definition 9** ([22]). *The intuitionistic fuzzy reasoning triple I model is denoted as*

$$\mathcal{R}_{L^*}(\mathcal{R}_{L^*}(A_{L^*}(x), B_{L^*}(y)), \mathcal{R}_{L^*}(A_{L^*}^*(x), B_{L^*}^*(y))) \tag{1}$$

*where $A_{L^*}, A_{L^*}^* \in IFS(X)$, $B_{L^*}, B_{L^*}^* \in IFS(Y)$, and $\mathcal{R}_{L^*}$ is the intuitionistic residuated implication on $L^*$. The smallest (greatest) intuitionistic fuzzy set $B_{L^*}^*(A_{L^*}^*)$ of the universe $Y(X)$ such that Formula (1) attains the greatest value is called the intuitionistic fuzzy reasoning triple I solution for FMP(FMT) problem.*

**Theorem 1** ([22]). *Suppose that $\mathcal{R}_{L^*}$ is the intuitionistic residuated implication induced by left-continuous associated intuitionistic t-norm $\mathcal{T}_{L^*}$, then*

(1) *The intuitionistic fuzzy reasoning triple I solution for FMP (IFMP algorithm solution $B_{L^*}^*$ for short) is given by the following formula*

$$B_{L^*}^*(y) = \sup_{x \in X} \mathcal{T}_{L^*}(A_{L^*}^*(x), \mathcal{R}_{L^*}(A_{L^*}(x), B_{L^*}(y))) \quad (\forall y \in Y). \tag{2}$$

(2) *The intuitionistic fuzzy reasoning triple I solution for FMT (IFMT algorithm solution $A_{L^*}^*$ for short) is given by the following formula*

$$A_{L^*}^*(x) = \inf_{y \in Y} \mathcal{R}_{L^*}(\mathcal{R}_{L^*}(A_{L^*}(x), B_{L^*}(y)), B_{L^*}^*(y)) \quad (\forall x \in X). \tag{3}$$

**Definition 10** ([14]). *The interval-valued fuzzy reasoning triple I model is denoted as*

$$\mathcal{R}_{L^I}(\mathcal{R}_{L^I}(A_{L^I}(x), B_{L^I}(y)), \mathcal{R}_{L^I}(A_{L^I}^*(x), B_{L^I}^*(y))) \tag{4}$$

*where $A_{L^I}, A_{L^I}^* \in IVFS(X)$, $B_{L^I}, B_{L^I}^* \in IVFS(Y)$, and $\mathcal{R}_{L^I}$ is the interval-valued residuated implication on $L^I$. The smallest (greatest) interval-valued fuzzy set $B_{L^I}^*(A_{L^I}^*)$ of the universe $Y(X)$ such that the Formula (4) attains the greatest value is called the interval-valued fuzzy reasoning triple I solution for FMP (FMT) problem.*

**Theorem 2** ([14]). *Suppose that $\mathcal{R}_{L^I}$ is the interval-valued residuated implication induced by left-continuous associated interval-valued t-norm $\mathcal{T}_{L^I}$, then*

(1) *The interval-valued fuzzy reasoning triple I solution for FMP (IVFMP algorithm solution $B_{L^I}^*$ for short) is given by the following formula*

$$B_{L^I}^*(y) = \sup_{x \in X} \mathcal{T}_{L^I}(\mathcal{R}_{L^I}(A_{L^I}(x), B_{L^I}(y)), A_{L^I}^*(x)) \quad (\forall y \in Y). \tag{5}$$

(2) *The interval-valued fuzzy reasoning triple I solution for FMT (IVFMT algorithm solution $A_{L^I}^*$ for short) is given by the following formula*

$$A_{L^I}^*(x) = \inf_{y \in Y} \mathcal{R}_{L^I}(\mathcal{R}_{L^I}(A_{L^I}(x), B_{L^I}(y)), B_{L^I}^*(y)) \quad (\forall x \in X). \tag{6}$$

**Theorem 3.** *The residuated lattice $(IFS(X), \cup_{L^*}, \cap_{L^*}, 0_*, 1_*, \mathcal{T}_{L^*}, \mathcal{R}_{L^*})$ and $(IVFS(X), \cup_{L^I}, \cap_{L^I}, 0_I, 1_I, \mathcal{T}_{L^I}, \mathcal{R}_{L^I})$ is isomorphic, where $\mathcal{R}_{L^*}$ is intuitionistic residuated implication induced by the left-continuous associated intuitionistic t-norm $\mathcal{T}_{L^*}$, $\mathcal{R}_{L^I}$ is interval-valued residuated implication induced by the left-continuous associated interval-valued t-norm $\mathcal{T}_{L^I}$.*

**Proof.** Let mapping $\varphi : IFS(X) \rightarrow IVFS(X)$, $(x_1, x_2) \mapsto [x_1, 1 - x_2]$, we prove that $\varphi$ is an isomorphism between the residuated lattice $(IFS(X), \cup_{L^*}, \cap_{L^*}, 0_*, 1_*, \mathcal{T}_{L^*}, \mathcal{R}_{L^*})$ and $(IVFS(X), \cup_{L^I}, \cap_{L^I}, 0_I, 1_I, \mathcal{T}_{L^I}, \mathcal{R}_{L^I})$. According to Lemma 3, we have $(IFS(X), \cup_{L^*}, \cap_{L^*}, 0_*, 1_*) \cong (IVFS(X), \cup_{L^I}, \cap_{L^I}, 0_I, 1_I)$.

Let $\alpha = (x_1, x_2), \beta = (y_1, y_2) \in L^*$, then

$$
\begin{aligned}
&\varphi(\mathcal{T}_{L^*}(\alpha, \beta)) \\
=\ &\varphi(\mathcal{T}_{L^*}((x_1, x_2), (y_1, y_2))) \\
=\ &\varphi(T((x_1, y_1), S(x_2, y_2))) \quad \text{(By Example 2)} \\
=\ &[T(x_1, y_1), 1 - S(x_2, y_2)] \\
=\ &[T(x_1, y_1), T(1 - x_2, 1 - y_2)] \\
=\ &\mathcal{T}_{L^I}([x_1, 1 - x_2], [y_1, 1 - y_2]) \quad \text{(By Example 3)} \\
=\ &\mathcal{T}_{L^I}(\varphi(x_1, x_2), \varphi(y_1, y_2)) \\
=\ &\mathcal{T}_{L^I}(\varphi(\alpha), \varphi(\beta))
\end{aligned}
$$

$$
\begin{aligned}
&\varphi(\mathcal{R}_{L^*}(\alpha, \beta)) \\
=\ &\varphi(\mathcal{R}_{L^*}((x_1, x_2), (y_1, y_2))) \\
=\ &\varphi(R(x_1, y_1) \wedge R(1 - x_2, 1 - y_2), 1 - R(1 - x_2, 1 - y_2)) \quad \text{(By Lemma 1)} \\
=\ &[R(x_1, y_1) \wedge R(1 - x_2, 1 - y_2), R(1 - x_2, 1 - y_2)] \\
=\ &\mathcal{R}_{L^I}([x_1, 1 - x_2], [y_1, 1 - y_2]) \quad \text{(By Lemma 2)} \\
=\ &\mathcal{R}_{L^I}(\varphi(x_1, x_2), \varphi(y_1, y_2)) \\
=\ &\mathcal{R}_{L^I}(\varphi(\alpha), \varphi(\beta))
\end{aligned}
$$

$\square$

**Theorem 4.** *There is a bijection between the IFMP algorithm solution $B_{L^*}^*$ (by Formula (2)) and the IVFMP algorithm solution $B_{L^I}^*$ (by Formula (5)).*

**Proof.** Let mapping $\varphi : IFS(Y) \to IVFS(Y), (y_1, y_2) \mapsto [y_1, 1 - y_2]$.

$$
\begin{aligned}
&\varphi(B_{L^*}^*(y)) \\
=\ &\varphi \sup_{x \in X} \mathcal{T}_{L^*}(A_{L^*}^*(x), \mathcal{R}_{L^*}(A_{L^*}(x), B_{L^*}(y))) \\
=\ &\sup_{x \in X} \varphi(\mathcal{T}_{L^*}(A_{L^*}^*(x), \mathcal{R}_{L^*}(A_{L^*}(x), B_{L^*}(y)))) \\
=\ &\sup_{x \in X} \mathcal{T}_{L^I}(\varphi(A_{L^*}^*(x)), \varphi(\mathcal{R}_{L^*}(A_{L^*}(x), B_{L^*}(y)))) \quad \text{(By Theorem 3)} \\
=\ &\sup_{x \in X} \mathcal{T}_{L^I}(\varphi(A_{L^*}^*(x)), \mathcal{R}_{L^I}(\varphi(A_{L^*}(x)), \varphi(B_{L^*}(y)))) \quad \text{(By Theorem 3)} \\
=\ &\sup_{x \in X} \mathcal{T}_{L^I}(A_{L^I}^*(x), \mathcal{R}_{L^I}(A_{L^I}(x), B_{L^I}(y))) \\
=\ &B_{L^I}^*(y)
\end{aligned}
$$

It is shown that there is a bijection between the IFMP algorithm solution $B_{L^*}^*$ and the IVFMP algorithm solution $B_{L^I}^*$. $\square$

**Theorem 5.** *There is a bijection between the IFMT algorithm solution $A_{L^*}^*$ (by Formula (3)) and the IVFMT algorithm solution $A_{L^I}^*$ (by Formula (6)).*

**Proof.** Let mapping $\varphi : IFS(X) \to IVFS(X), (x_1, x_2) \mapsto [x_1, 1 - x_2]$.

$$
\begin{aligned}
&\varphi(A_{L^*}^*(x)) \\
=\ &\varphi \inf_{y \in Y} \mathcal{R}_{L^*}(\mathcal{R}_{L^*}(A_{L^*}(x), B_{L^*}(y)), B_{L^*}^*(y)) \\
=\ &\inf_{y \in Y} \varphi(\mathcal{R}_{L^*}(\mathcal{R}_{L^*}(A_{L^*}(x), B_{L^*}(y)), B_{L^*}^*(y))) \\
=\ &\inf_{y \in Y} \mathcal{R}_{L^I}(\varphi(\mathcal{R}_{L^*}(A_{L^*}(x), B_{L^*}(y))), \varphi(B_{L^*}^*(y))) \quad \text{(By Theorem 3)} \\
=\ &\inf_{y \in Y} \mathcal{R}_{L^I}(\varphi(\mathcal{R}_{L^*}(A_{L^*}(x), B_{L^*}(y))), B_{L^I}^*(y)) \quad \text{(By Theorem 3)} \\
=\ &\inf_{y \in Y} \mathcal{R}_{L^I}(\mathcal{R}_{L^I}(A_{L^I}(x), B_{L^I}(y)), B_{L^I}^*(y)) \\
=\ &A_{L^I}^*(x)
\end{aligned}
$$

It is shown that there is a bijection between the IFMT algorithm solution $A_{L^*}^*$ and the IVFMT algorithm solution $A_{L^I}^*$. $\square$

**Example 4.** *The intuitionistic fuzzy numbers $A_{L^*}, A_{L^*}^*, B_{L^*}$ are shown in Table 1, and the intuitionistic fuzzy numbers $A, A^*, B$ is transformed into interval-valued fuzzy numbers $A_{L^I}, A_{L^I}^*, B_{L^I}$ by mapping $\varphi$, as can be seen in Table 2. Take the triangular norm $T = T_G$, the intuitionistic fuzzy*

*reasoning triple I solutions $B_{L^*}^*$ and the interval-valued fuzzy reasoning triple I solutions $B_{L^I}^*$ are shown in Table 3.*

**Table 1.** Data of $A_{L^*}, A_{L^*}^*$ and $B_{L^*}$.

|  | $x_1$ | $x_2$ | $x_3$ |
|---|---|---|---|
| $A_{L^*}$ | $(0.60, 0.30)$ | $(0.90, 0.10)$ | $(0.40, 0.50)$ |
| $A_{L^*}^*$ | $(0.20, 0.50)$ | $(0.30, 0.60)$ | $(0.10, 0.40)$ |
|  | $y_1$ | $y_2$ | $y_3$ |
| $B_{L^*}$ | $(0.50, 0.30)$ | $(0.30, 0.60)$ | $(0.10, 0.70)$ |

**Table 2.** Data of $A_{L^I}, A_{L^I}^*$ and $B_{L^I}$.

|  | $x_1$ | $x_2$ | $x_3$ |
|---|---|---|---|
| $A_{L^I}$ | $[0.60, 0.70]$ | $[0.90, 0.90]$ | $[0.40, 0.50]$ |
| $A_{L^I}^*$ | $[0.20, 0.50]$ | $[0.30, 0.40]$ | $[0.10, 0.60]$ |
|  | $y_1$ | $y_2$ | $y_3$ |
| $B_{L^I}$ | $[0.50, 0.70]$ | $[0.30, 0.40]$ | $[0.10, 0.30]$ |

**Table 3.** IFMP algorithm solutions $B_{L^*}^*$ and IVFMP algorithm solutions $B_{L^I}^*$.

|  | $y_1$ | $y_2$ | $y_3$ |
|---|---|---|---|
| $B_{L^*}^*$ | $(0.30, 0.40)$ | $(0.30, 0.60)$ | $(0.10, 0.70)$ |
| $B_{L^I}^*$ | $[0.30, 0.60]$ | $[0.30, 0.40]$ | $[0.10, 0.30]$ |

Use the mapping $\varphi(B_{L^*}^*) = D$, the calculation results are shown in Table 4. By comparing the data in Tables 3 and 4, the value of $D$ is equal to the solution $B_{L^I}^*$ for solving the IVFMP problem. The results show that the solutions based on the two fuzzy sets are in one-to-one correspondence.

**Table 4.** The values of corresponding to under the mapping $\varphi$.

|  |  |  |  |
|---|---|---|---|
| $B_{L^*}^*$ | $(0.30, 0.40)$ | $(0.30, 0.60)$ | $(0.10, 0.70)$ |
| $D$ | $[0.30, 0.60]$ | $[0.30, 0.40]$ | $[0.10, 0.30]$ |

*3.2. The Relationship between the Reverse Triple I Methods Based on Intuitionistic Fuzzy Sets and Interval-Valued Fuzzy Sets*

In this section, the solutions of the reverse triple I method based on the IFS and the IVFS have been given, and the relationship between the two solutions will be studied.

**Definition 11** ([24])**.** *The intuitionistic fuzzy reasoning reverse triple I model is denoted as*

$$\mathcal{R}_{L^*}(\mathcal{R}_{L^*}(A_{L^*}^*(x), B_{L^*}^*(y)), \mathcal{R}_{L^*}(A_{L^*}(x), B_{L^*}(y))) \tag{7}$$

*where $A_{L^*}, A_{L^*}^* \in IFS(X)$, $B_{L^*}, B_{L^*}^* \in IFS(Y)$, and $\mathcal{R}_{L^*}$ is the intuitionistic residuated implication on $L^*$. The greatest (smallest) intuitionistic fuzzy set $B_{L^*}^*(A_{L^*}^*)$ of the universe $Y(X)$ such that the Formula (7) attains the greatest value is called the intuitionistic fuzzy reasoning reverse triple I solution for FMP (FMT) problem, denoted by $B_{RL^*}^*(A_{RL^*}^*)$.*

**Theorem 6** ([24])**.** *Let $\mathcal{R}_{L^*}$ be the intuitionistic residuated implication induced by left-continuous associated intuitionistic t-norm $\mathcal{T}_{L^*}$. Then*

*(1)   The intuitionistic fuzzy reasoning reverse triple I solution for FMP is given by the following formula*

$$B^*_{RL^*}(y) = \inf_{x \in X} \mathcal{T}_{L^*}(A^*_{L^*}(x), \mathcal{R}_{L^*}(A_{L^*}(x), B_{L^*}(y))) \quad (\forall y \in Y). \tag{8}$$

*(2)   The intuitionistic fuzzy reasoning reverse triple I solution for FMT is given by the following formula*

$$A^*_{RL^*}(x) = \sup_{y \in Y} \mathcal{R}_{L^*}(\mathcal{R}_{L^*}(A_{L^*}(x), B_{L^*}(y)), B^*_{L^*}(y)) \quad (\forall x \in X). \tag{9}$$

**Definition 12** ([17]). *The interval-valued fuzzy reasoning reverse triple I model is denoted as*

$$\mathcal{R}_{L^I}(\mathcal{R}_{L^I}(A^*_{L^I}(x), B^*_{L^I}(y)), \mathcal{R}_{L^I}(A_{L^I}(x), B_{L^I}(y))) \tag{10}$$

*where $A_{L^I}(x), A^*_{L^I}(x) \in IVFS(X)$, $B_{L^I}(y), B^*_{L^I}(y) \in IVFS(Y)$, and $\mathcal{R}_{L^I}$ is the interval-valued residuated implication on $L^I$. The greatest (smallest) interval-valued fuzzy set $B^*_{L^I}(A^*_{L^I})$ of the universe $Y(X)$ such that the Formula (10) attains the greatest value is called the interval-valued fuzzy reasoning reverse triple I solution for the FMP (FMT) problem, denoted by $B^*_{RL^I}(A^*_{RL^I})$.*

**Theorem 7** ([17]). *Let $\mathcal{R}_{L^I}$ be the interval-valued residuated implication induced by left-continuous associated interval-valued t-norm $\mathcal{T}_{L^I}$. Then*

*(1)   The interval-valued fuzzy reasoning reverse triple I solution for FMP is given by the following formula*

$$B^*_{RL^I}(y) = \inf_{x \in X} \mathcal{T}_{L^I}(\mathcal{R}_{L^I}(A_{L^I}(x), B_{L^I}(y)), A^*_{L^I}(x)) \quad (\forall y \in Y). \tag{11}$$

*(2)   The interval-valued fuzzy reasoning reverse triple I solution for FMT is given by the following formula*

$$A^*_{RL^I}(x) = \sup_{y \in Y} \mathcal{R}_{L^I}(\mathcal{R}_{L^I}(A_{L^I}(x), B_{L^I}(y)), B^*_{L^I}(y)) \quad (\forall x \in X). \tag{12}$$

**Theorem 8.** *There is a bijection between the intuitionistic fuzzy reasoning reverse triple I solution $B^*_{RL^*}$ for FMP (by Formula (8)) and the interval-valued fuzzy reasoning reverse triple I solution $B^*_{RL^I}$ for FMP (by Formula (11)).*

**Proof.** Let mapping $\varphi : IFS(Y) \to IVFS(Y)$, $(y_1, y_2) \mapsto [y_1, 1 - y_2]$.

$$
\begin{aligned}
& \varphi(B^*_{RL^*}(y)) \\
=\ & \varphi \inf_{x \in X} \mathcal{T}_{L^*}(A^*_{L^*}(x), \mathcal{R}_{L^*}(A_{L^*}(x), B_{L^*}(y))) \\
=\ & \inf_{x \in X} \varphi(\mathcal{T}_{L^*}(A^*_{L^*}(x), \mathcal{R}_{L^*}(A_{L^*}(x), B_{L^*}(y)))) \\
=\ & \inf_{x \in X} \mathcal{T}_{L^I}(\varphi(A^*_{L^*}(x)), \varphi(\mathcal{R}_{L^*}(A_{L^*}(x), B_{L^*}(y)))) \quad \text{(By Theorem 3)} \\
=\ & \inf_{x \in X} \mathcal{T}_{L^I}(\varphi(A^*_{L^*}(x)), \mathcal{R}_{L^I}(\varphi(A_{L^*}(x)), \varphi(B_{L^*}(y)))) \quad \text{(By Theorem 3)} \\
=\ & \inf_{x \in X} \mathcal{T}_{L^I}(A^*_{L^I}(x), \mathcal{R}_{L^I}(A_{L^I}(x), B_{L^I}(y))) \\
=\ & B^*_{RL^I}(y)
\end{aligned}
$$

It is shown that there is a bijection between the intuitionistic fuzzy reasoning reverse triple I solution $B^*_{RL^*}$ and the interval-valued fuzzy reasoning reverse triple I solution $B^*_{RL^I}$.  □

**Theorem 9.** *There is a bijection between the intuitionistic fuzzy reasoning reverse triple I solution $A^*_{RL^*}$ for FMT (by Formula (9)) and the interval-valued fuzzy reasoning reverse triple I solution $A^*_{RL^I}$ for FMT (by Formula (12)).*

**Proof.** Let mapping $\varphi : IFS(X) \to IVFS(X)$, $(x_1, x_2) \mapsto [x_1, 1 - x_2]$.

$$
\begin{aligned}
& \varphi(A^*_{RL^*}(x)) \\
=\ & \varphi \inf_{y \in Y} (\mathcal{R}_{L^*}(\mathcal{R}_{L^*}(A_{L^*}(x), B_{L^*}(y)), B^*_{L^*}(y))) \\
=\ & \inf_{y \in Y} \varphi(\mathcal{R}_{L^*}(\mathcal{R}_{L^*}(A_{L^*}(x), B_{L^*}(y)), B^*_{L^*}(y))) \quad \text{(By Lemma 3)} \\
=\ & \inf_{y \in Y} \mathcal{R}_{L^I}(\varphi(\mathcal{R}_{L^*}(A_{L^*}(x), B_{L^*}(y))), \varphi(B^*_{L^*}(y))) \quad \text{(By Theorem 3)} \\
=\ & \inf_{y \in Y} \mathcal{R}_{L^I}(\varphi(\mathcal{R}_{L^*}(A_{L^*}(x), B_{L^*}(y))), B^*_{L^I}(y)) \quad \text{(By Theorem 3)} \\
=\ & \inf_{y \in Y} \mathcal{R}_{L^I}(\mathcal{R}_{L^I}(A_{L^I}(x), B_{L^I}(y)), B^*_{L^I}(y)) \\
=\ & A^*_{RL^I}(x)
\end{aligned}
$$

It is shown that there is a bijection between the intuitionistic fuzzy reasoning reverse triple I solution $A^*_{RL^*}$ and the interval-valued fuzzy reasoning reverse triple I solution $A^*_{RL^I}$. $\square$

*3.3. The Relationship between the SIS Methods Based on Intuitionistic Fuzzy Sets and Interval-Valued Fuzzy Sets*

In this section, the solutions of the SIS method based on the IFS and the IVFS have been given, and the relationship between the two solutions will be studied.

**Definition 13** ([25]). *Let $A, B \in IFS(X)$, and $\mathcal{R}_{L^*}$ be the intuitionistic residuated implication induced by left-continuous associated intuitionistic t-norm $\mathcal{T}_{L^*}$. Then, the intuitionistic fuzzy reasoning subsethood degree $S_{L^*}$ is denoted as*

$$
S_{L^*}(A, B) = \inf_{x \in X} (A(x), B(x))
$$

**Definition 14** ([25]). *The intuitionistic fuzzy reasoning SIS model is denoted as*

$$
\mathcal{R}_{L^*}(S_{L^*}(A^*_{L^*}, A_{L^*}), S_{L^*}(B^*_{L^*}, B_{L^*})) \tag{13}
$$

*where $A_{L^*}, A^*_{L^*} \in IFS(X)$, $B_{L^*}, B^*_{L^*} \in IFS(Y)$, and $\mathcal{R}_{L^*}$ is the intuitionistic residuated implication on $L^*$. The greatest intuitionistic fuzzy set $B^*_{L^*}(A^*_{L^*})$ of the universe $Y(X)$ such that the Formula (13) attains the greatest value is called the intuitionistic fuzzy reasoning SIS algorithm solution for FMP(FMT) problem, denoted by $B^*_{SL^*}(A^*_{SL^*})$.*

**Theorem 10** ([25]). *Let $\mathcal{R}_{L^*}$ be the intuitionistic residuated implication induced by left-continuous associated intuitionistic t-norm $\mathcal{T}_{L^*}$. Then:*

*(1) The intuitionistic fuzzy reasoning SIS reasoning algorithm solution for FMP is given by the following formula*

$$
B^*_{SL^*}(y) = \inf_{x \in X} \mathcal{R}_{L^*}(S_{L^*}(A^*_{L^*}, A_{L^*}), B_{L^*}(y)) \quad (\forall y \in Y). \tag{14}
$$

*(2) The intuitionistic fuzzy reasoning SIS reasoning algorithm solution for FMT is given by the following formula*

$$
A^*_{SL^*}(x) = \inf_{y \in Y} \mathcal{R}_{L^*}(S_{L^*}(B^*_{L^*}, B_{L^*}), A_{L^*}(x)) \quad (\forall x \in X). \tag{15}
$$

**Definition 15** ([18])**.** *Let* $A, B \in IVFS(X)$, *and* $\mathcal{R}_{L^I}$ *be the interval-valued residuated implication induced by left-continuous associated interval-valued t-norm* $\mathcal{T}_{L^I}$. *Then, the interval-valued fuzzy reasoning subsethood degree* $S_{L^I}$ *is denoted as*

$$S_{L^I}(A, B) = \inf_{x \in X} (A(x), B(x))$$

**Definition 16** ([18])**.** *The interval-valued fuzzy reasoning SIS algorithm model is denoted as*

$$\mathcal{R}_{L^I}(S_{L^I}(B_{L^I}^*, B_{L^I}), S_{L^I}(A_{L^I}^*, A_{L^I})) \tag{16}$$

*where* $A_{L^I}(x), A_{L^I}^*(x) \in IVFS(X)$, $B_{L^I}(y), B_{L^I}^*(y) \in IVFS(Y)$, *and* $\mathcal{R}_{L^I}$ *is the interval-valued residuated implication on* $L^I$. *The greatest interval-valued fuzzy set* $B_{L^I}^*(A_{L^I}^*)$ *of the universe* $Y(X)$ *such that the Formula (16) attains the greatest value is called the interval-valued fuzzy reasoning SIS algorithm solution for the FMP(FMT) problem, denoted by* $B_{SL^I}^*(A_{SL^I}^*)$.

**Theorem 11** ([18])**.** *Let* $\mathcal{R}_{L^I}$ *be the interval-valued residuated implication induced by left-continuous associated interval-valued t-norm* $\mathcal{T}_{L^I}$. *Then:*

*(1)* *The interval-valued fuzzy reasoning SIS algorithm solution for FMP is given by the following formula*

$$B_{SL^I}^*(y) = \inf_{x \in X} \mathcal{R}_{L^I}(S_{L^I}(A_{L^I}^*, A_{L^I}), B_{L^I}(y)) \quad (\forall y \in Y). \tag{17}$$

*(2)* *The interval-valued fuzzy reasoning SIS algorithm solution for FMT is given by the following formula*

$$A_{SL^I}^*(x) = \inf_{y \in Y} \mathcal{R}_{L^I}(S_{L^I}(B_{L^I}^*, B_{L^I}), A_{L^I}(x)) \quad (\forall x \in X). \tag{18}$$

**Theorem 12.** *There is a bijection between the intuitionistic fuzzy reasoning SIS algorithm solution* $B_{SL^*}^*$ *for FMP (by Formula (14)) and the interval-valued fuzzy reasoning SIS algorithm solution* $B_{SL^I}^*$ *for FMP (by Formula (17)).*

**Proof.** Let mapping $\varphi : IFS(Y) \to IVFS(Y)$, $(y_1, y_2) \mapsto [y_1, 1 - y_2]$.

$$
\begin{aligned}
& \varphi(B_{SL^*}^*(y)) \\
= & \ \varphi \inf_{x \in X} (\mathcal{R}_{L^*}(S_{L^*}(A_{L^*}^*, A_{L^*}), B_{L^*}(y))) \\
= & \ \inf_{x \in X} \varphi(\mathcal{R}_{L^*}(S_{L^*}(A_{L^*}^*, A_{L^*}), B_{L^*}(y))) \\
= & \ \inf_{x \in X} \mathcal{R}_{L^I}(\varphi(S_{L^*}(A_{L^*}^*, A_{L^*}), B_{L^*}(y))) \\
= & \ \inf_{x \in X} \mathcal{R}_{L^I}(S_{L^I}(\varphi((A_{L^*}^*, A_{L^*}), \varphi(B_{L^*}(y)))) \\
= & \ \inf_{x \in X} (\mathcal{R}_{L^I}(S_{L^I}(A_{L^I}^*, A_{L^I}), B_{L^I}(y))) \\
= & \ B_{SL^I}^*(y)
\end{aligned}
$$

It is shown that there is a bijection between the intuitionistic fuzzy reasoning SIS algorithm solution $B_{SL^*}^*$ and the interval-valued fuzzy reasoning SIS algorithm solution $B_{SL^I}^*$. $\square$

**Theorem 13.** *There is a bijection between the intuitionistic fuzzy reasoning SIS algorithm solution* $A_{SL^*}^*$ *for FMT (by Formula (15)) and the interval-valued fuzzy reasoning SIS algorithm solution* $A_{SL^I}^*$ *for FMT (by Formula (18)).*

**Proof.** Let mapping $\varphi : IFS(X) \rightarrow IVFS(X)$, $(x_1, x_2) \mapsto [x_1, 1 - x_2]$.

$$
\begin{aligned}
& \varphi(A_{SL^*}^*(x)) \\
=\ & \varphi \inf_{y \in Y} (\mathcal{R}_{L^*}(S_{L^*}(B_{L^*}^*, B_{L^*})A_{L^*}(x)) \\
=\ & \inf_{y \in Y} \varphi(\mathcal{R}_{L^*}(S_{L^*}(B_{L^*}^*, B_{L^*}), A_{L^*}(x))) \\
=\ & \inf_{y \in Y} \mathcal{R}_{L^I}(\varphi(S_{L^*}(B_{L^*}^*, B_{L^*}), A_{L^*}(x))) \\
=\ & \inf_{y \in Y} \mathcal{R}_{L^I}(S_{L^I}(\varphi((B_{L^*}^*, B_{L^*}), \varphi(A_{L^*}(x))))) \\
=\ & \inf_{y \in Y} (\mathcal{R}_{L^I}(S_{L^I}(B_{L^I}^*, B_{L^I}), A_{L^I}(x))) \\
=\ & A_{SL^I}^*(x)
\end{aligned}
$$

It is shown that there is a bijection between the intuitionistic fuzzy reasoning SIS algorithm solution $A_{SL^*}^*$ and the interval-valued fuzzy reasoning SIS algorithm solution $A_{SL^I}^*$. $\square$

The flow diagram of results is shown in Figure 1.

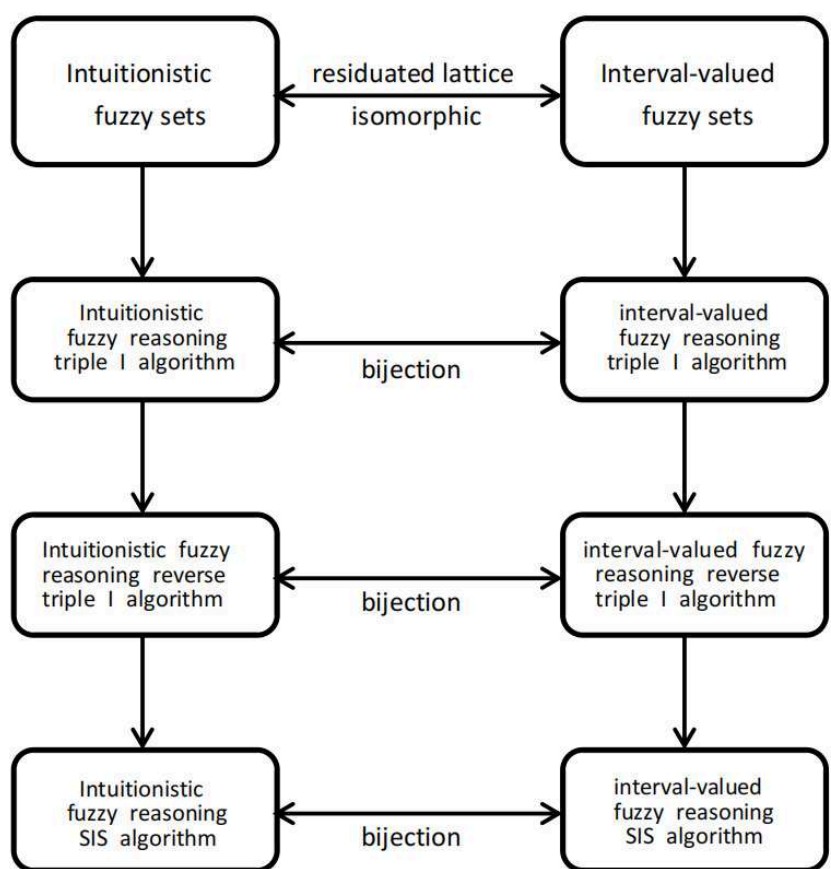

**Figure 1.** The flow diagram of results.

## 4. Conclusions

In this paper, we studied the relationship between intuitionistic fuzzy reasoning algorithm and interval-valued fuzzy reasoning algorithm. It is proved that there is a bijection between the intuitionistic fuzzy reasoning triple I solution and the interval-valued fuzzy reasoning triple I solution, and there is a bijection between the intuitionistic fuzzy reasoning reverse triple I solution and the interval-valued fuzzy reasoning reverse triple I solution. Moreover, it is proved that there is a bijection between the intuitionistic fuzzy reasoning

SIS solution and the interval-valued fuzzy reasoning SIS solution. Finally, a numerical example is given to show that there is a bijection between the intuitionistic fuzzy reasoning triple I method and the interval-valued fuzzy reasoning triple I method. We prove that the intuitionistic fuzzy reasoning method and interval-valued fuzzy reasoning method are equivalent in essence. In practical application, interval-valued fuzzy sets can effectively reduce the loss of fuzzy information, and intuitionistic fuzzy sets can characterize information from two aspects, intuitionistic fuzzy reasoning method and interval-valued fuzzy reasoning method can be used for one calculation and one test. Intuitionistic fuzzy reasoning method and interval-valued fuzzy reasoning method will be applied in many fields such as pattern recognition and medical diagnosis. In the future, how to apply the algorithm to practical applications is the next research direction. we will study how to apply intuitionistic fuzzy reasoning method and an interval-valued fuzzy reasoning method to practical problems such as pattern recognition and medical diagnosis.

**Author Contributions:** M.L. initiated the research and provided the framework of this paper. W.L. and H.S. wrote and completed this paper with M.L. who validated the findings and provided helpful suggestions. All authors have read and agreed to the published version of the manuscript.

**Funding:** This work is supported by the National Natural Science Foundation of China (No. 12171445).

**Institutional Review Board Statement:** Not applicable.

**Informed Consent Statement:** Informed consent was obtained from all subjects involved in the study.

**Data Availability Statement:** Not applicable.

**Conflicts of Interest:** The authors declare no conflict of interest.

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
