# Peer review of "The Relationship between Fuzzy Reasoning Methods Based on Intuitionistic Fuzzy Sets and Interval-Valued Fuzzy Sets"

_axioms, doi:10.3390/axioms11080419_

Round 1
Reviewer 1 Report
The papers motivation is good but it not written well. I recommended it for major revision:
(1) There is very tough to identify from the mathematical equation that which is developed and which is already taken from others. I suggest you to mark briefly after every new findings.
(2) A section should be dedicated for numerical example. Where several numerical analysis should be perform with sensitivity analysis.
(3) Give a flow diagram that how your papers results are shown or structured in a scientific manner.
(4) Is there any scientific application have ? If so then explain briefly.
(5) Fuzzy sets results should also compare with IVIF and IFS.
(7) Important equation should be numbered and should cite in the text.
(8) Future research should be written in brief in conclusion section.
(9) The grammatical mistakes should be corrected.
Reviewer 2 Report
The paper contains an interesting (new) view on the relationship between the intuitionistic fuzzy sets (IFSs) and interval-valued fuzzy sets (IVFSs). But the theory of interval-valued intuitionistic fuzzy sets (IVIFSs) was extended essentially in
Atanassov, K. Interval-Valued Intuitionistic Fuzzy Sets, Springer, Cham, 2020
and this fact must be discussed. In this book another form of representation of IVIFSs by IFSs is discussed and it will be interesting if the authors study and represent it, too (probably, in a next research of theirs). Also, the mapping from Lemma 3 is given in [2], i.e. 14 years before [8].
Reviewer 3 Report
The paper studies the full implication triple I method, the reverse triple I method, and the SIS algorithm for intuitionistic fuzzy sets and interval-valued fuzzy sets. The main results of the paper are Theorems 4, 5, 8, 9, 12, and 13. Were similar bijections described by other authors?I noticed only one misprint in line 171, there is no lemma 2.3 in the paper. But I have two questions more. Does an interval-valued fuzzy set coincide with an interval type 2 fuzzy set? If this is true, is it possible to generalize the methods for arbitrary type 2 fuzzy sets?
Reviewer 4 Report
Dear Authors and Editors:
The reviewer would like to send you the review results report.
Please find the attached file in this email.
Thank you
Sincerely yours,
The reviewer

Round 2
Reviewer 1 Report
The paper is now acceptable.